# Citrulline Plus Arginine Induces an Angiogenic Response and Increases Permeability in Retinal Endothelial Cells via Nitric Oxide Production

**DOI:** 10.3390/ijms26052080

**Published:** 2025-02-27

**Authors:** Cassandra Warden, Daniella Zubieta, Milam A. Brantley

**Affiliations:** Department of Ophthalmology and Visual Sciences, Vanderbilt Eye Institute, Vanderbilt University Medical Center, Nashville, TN 37232, USA; cwarden@oakland.edu (C.W.); daniella.zubieta@vumc.org (D.Z.)

**Keywords:** arginine, citrulline, angiogenesis, retinal endothelial cell, vascular permeability, arginase, endothelial nitric oxide synthase, nitric oxide

## Abstract

We previously observed elevated plasma levels of citrulline and arginine in diabetic retinopathy patients compared to diabetic controls. We tested our hypothesis that citrulline plus arginine induces angiogenesis and increases permeability in retinal endothelial cells. Human retinal microvascular endothelial cells (HRMECs) were treated with citrulline, arginine, or citrulline + arginine, and angiogenesis was measured with cell proliferation, migration, and tube formation assays. Permeability was measured in HRMEC monolayers via trans-endothelial electrical resistance (TEER) and FITC-labeled dextran. We also measured arginase activity, arginase-1 and arginase-2 expression, protein expression and phosphorylation of endothelial nitric oxide synthase (eNOS), and nitric oxide (NO) production. Citrulline + arginine induced endothelial cell proliferation (*p* = 0.018), migration (*p* = 0.011), and tube formation (*p* = 0.0042). Citrulline + arginine also increased FITC-dextran flow-through (*p* = 1.5 × 10^−5^) and decreased TEER (*p* = 0.010). Citrulline + arginine had no effect on arginase activity, but it increased eNOS (*p* = 6.3 × 10^−4^) and phosphorylated eNOS (*p* = 0.029), as well as NO production (*p* = 0.025). Inhibiting eNOS prevented the increase in NO (*p* = 0.0092), inhibited citrulline + arginine-induced cell migration (*p* = 0.0080) and tube formation (*p* = 0.0092), and blocked citrulline + arginine-related alterations in FITC-dextran flow-through (*p* = 3.6 × 10^−4^) and TEER (*p* = 3.9 × 10^−4^). These data suggest that citrulline + arginine treatment induces angiogenesis and increases permeability in retinal endothelial cells by activating eNOS and increasing NO production.

## 1. Introduction

Diabetic retinopathy (DR) is a major complication of diabetes and a leading cause of blindness worldwide [1]. DR can progress from non-proliferative DR (NPDR), which includes retinal hemorrhages and microaneurysms, to a proliferative stage (PDR), which may lead to vitreous hemorrhage and severe vision loss [2]. Predicting development and progression of DR is often challenging. Poor glycemic control and longer diabetes duration are strongly associated risk factors for DR [3] but do not fully explain the variability in clinical progression [4]. Treatments for PDR and diabetic macular edema (DME) include intravitreal injections of anti-vascular endothelial growth factor (VEGF) medication and laser photocoagulation, both of which are administered only after retinal edema or neovascularization is detected. It is thus important to continue the search for earlier interventions to prevent these vision-threatening complications of DR.

In a recent metabolome-wide association study of DR, our laboratory found elevated plasma levels of citrulline and arginine in patients with DR compared to patients with diabetes and no retinopathy [5]. Metabolic pathway analysis indicated that the urea cycle was altered in DR patients compared to diabetic controls [5]. In a later study, we used tandem mass spectrometry to quantitate plasma levels of six urea cycle metabolites (arginine, citrulline, argininosuccinic acid, ornithine, proline, and asymmetric dimethylarginine) in an expanded patient cohort and found that only citrulline and arginine were elevated in patients with DR compared to diabetic controls [6]. Other groups have reported elevated citrulline and arginine levels in the vitreous of PDR patients compared to non-diabetic controls [7,8], as well as elevated serum arginine levels in patients with severe DR compared to non-diabetic controls [9]. Since the primary causes of vision loss in DR are related to aberrant angiogenesis (PDR) and increased vascular permeability (DME), we sought to determine if exogenous citrulline and arginine individually or together could induce an angiogenic response or increase permeability in human retinal endothelial cells.

Arginine can serve as a substrate for two different enzymes in overlapping pathways: arginase, which produces urea as a product, and nitric oxide synthase (NOS), which produces nitric oxide (NO) and citrulline. Increased arginase activity has been associated with vascular changes in both diabetes and DR [10,11] and upregulation of endothelial NOS (eNOS) and elevated NO have been linked to DR and retinal neovascularization [12,13]. Therefore, we sought to determine if exogenous citrulline + arginine upregulates the arginase pathway or if their combination activates the citrulline-NO cycle through eNOS to produce nitric oxide.

In the current study, we show that citrulline + arginine induces angiogenesis and increases the permeability of retinal endothelial cells by activating eNOS and increasing NO production. This is an important first step in determining if these metabolites could serve as targets for the prevention or treatment of retinal vascular diseases.

## 2. Results

### 2.1. Citrulline Plus Arginine Induces an Angiogenic Response in HRMECs

Cell proliferation, cell migration, and tube formation assays were performed to measure the angiogenic response of primary human retinal microvascular endothelial cells (HRMECs) treated with citrulline, arginine, or citrulline + arginine. Cell proliferation was measured using a BrdU assay (Figure 1). Cells treated with vascular endothelial growth factor (VEGF) served as a positive control and displayed significantly increased proliferation compared to untreated control cells (45.0 % increase, *p* = 0.0061). Neither citrulline nor arginine alone induced cell proliferation (*p* = 0.064 and *p* = 0.22, respectively), but citrulline + arginine significantly induced cell proliferation compared to untreated controls (39.6% increase, *p* = 0.018).

To measure the effects of citrulline and arginine on cell migration, HRMECs were treated with citrulline, arginine, or citrulline + arginine, and a scratch wound assay was performed (Figure 2). HREMCs treated with VEGF induced cell migration compared to untreated controls (75.0% increase, *p* = 7.3 × 10^−4^), serving as a positive control. Neither citrulline nor arginine alone affected cell migration (*p* = 0.48 and *p* = 0.20, respectively), but citrulline + arginine stimulated cell migration compared to untreated controls (57.7% increase, *p* = 0.011).

To measure the effects of citrulline and arginine on endothelial tube formation, HRMECs were seeded onto growth reduced Matrigel with citrulline, arginine, or citrulline + arginine to quantify tube length (Figure 3). Cells treated with VEGF served as a positive control and produced significantly longer tubes compared to untreated cells (46.7% increase, *p* = 0.0019). Citrulline (*p* = 0.35) or arginine (*p* = 0.20) alone did not increase tube length compared to untreated controls. However, citrulline + arginine induced significant tube formation in HRMECs compared to untreated controls (35.6% increase, *p* = 0.0057). Together, these data show that citrulline + arginine can induce an angiogenic response in human retinal endothelial cells in vitro.

### 2.2. Citrulline Plus Arginine Increases HRMEC Monolayer Permeability

FITC-labeled dextran and trans-endothelial electrical resistance (TEER) assays were used to investigate changes to the HRMEC monolayer when treated with citrulline, arginine, or citrulline + arginine. HRMECs were grown on Transwell inserts, and FITC-dextran flow-through was used to measure changes in barrier function (Figure 4A). Cells treated with VEGF served as a positive control and resulted in higher fluorescence compared to untreated controls (43.7% increase, *p* = 0.022), indicating more FITC-dextran passed through the monolayer due to increased permeability. Citrulline (*p* = 1.0) or arginine (*p* = 1.0) alone did not increase fluorescence compared to untreated controls. Citrulline + arginine caused significantly increased fluorescence compared to the untreated HRMEC monolayer control (82.4% increase, *p* = 1.5 × 10^−5^), suggesting enhanced permeability.

TEER was measured in HRMEC monolayers treated with citrulline, arginine, or citrulline + arginine (Figure 4B). HRMECs treated with VEGF served as a positive control and resulted in a lower TEER reading 48 h after treatment administration compared to untreated controls (20.2% decrease, *p* = 0.0058), indicating tight junction disruption in the HRMEC monolayer. Citrulline (*p* = 1.0) or arginine (*p* = 0.73) alone caused no significant changes in TEER measurements compared to untreated controls. Citrulline + arginine caused significantly lower TEER measurements at 48 h after treatment compared to untreated controls (18.5% decrease, *p* = 0.010), suggesting tight junction disruption. Together, these data suggest that citrulline and arginine together, but not independently, can cause an increase in permeability of HRMEC monolayers.

### 2.3. Citrulline Plus Arginine Does Not Alter Arginase Activity in HRMECs

Arginine is a substrate for enzymes of two overlapping pathways: arginase of the urea cycle and eNOS of the citrulline-NO cycle. We measured the products of both enzymes to investigate which pathway is primarily stimulated in HRMECs in the presence of citrulline + arginine. To determine if arginase activity is altered in HRMECs treated with citrulline + arginine, we used a commercially available kit to measure total arginase activity (Figure 5A). Compared to controls, citrulline (*p* = 0.80), arginine (*p* = 0.31), and citrulline + arginine (*p* = 0.54) had no effect on total arginase activity.

Arginase has two isoforms, arginase-1, a cytosolic enzyme, and arginase-2, a mitochondrial enzyme, and while both use arginine as a substrate in the urea cycle, these isoforms can be induced by different stimuli [14]. To further investigate arginase response to citrulline + arginine in HRMECs, protein expression of arginase-1 and arginase-2 was measured using Western blots (Figure 5B–D). Compared to untreated controls, there was no difference in arginase-1 protein expression in HRMECs treated with citrulline (*p* = 0.64) or arginine (*p* = 1.0), but the addition of citrulline + arginine resulted in lower arginase-1 protein expression (52.3% decrease, *p* = 0.0046). No difference was observed in protein expression of arginase-2 in HRMECs treated with citrulline (*p* = 0.067), arginine (*p* = 0.61), or citrulline + arginine (*p* = 0.95). Thus, while citrulline + arginine led to a lower protein expression of arginase-1, total arginase activity was not affected.

### 2.4. Citrulline Plus Arginine Induces Intracellular NO Production Through eNOS Activation

NO production was measured using DAF-FM DA to stain NO in HRMECs, and fluorescent staining was quantified (Figure 6A,B). HRMECs treated with VEGF were used as a positive control and displayed significantly increased fluorescent intensity compared to untreated cells (374.0% increase, *p* = 6.7 × 10^−5^), indicating an increase in NO production. Cells treated with citrulline (235.9% increase, *p* = 0.012), arginine (274.7% increase, *p* = 0.0029), and citrulline + arginine (215.8% increase, *p* = 0.025) enhanced NO production compared to untreated controls. Increased production of the eNOS product, NO, suggests citrulline and arginine, alone and in combination, activate eNOS in HRMECs.

Phosphorylation sites of eNOS play a key role in activating or inhibiting the enzyme’s activity. Serine 1177 is a known activation site and is phosphorylated in the VEGF downstream signaling pathway [15]. To further investigate if citrulline and arginine activate eNOS in retinal endothelial cells, HRMECs were treated with citrulline, arginine, or citrulline + arginine, and Western blots were used to measure the level of total eNOS and eNOS phosphorylated at Serine 1177 (Figure 6C–E). HRMECs treated with VEGF served as a positive control, and VEGF treatment increased total eNOS (228.4% increase, *p* = 9.2 × 10^−4^) and phosphorylated eNOS (p-eNOS) protein expression (303.1% increase, *p* = 0.010) compared to untreated controls. Citrulline alone (*p* = 0.55) did not increase eNOS expression compared to controls, but arginine alone (158.6% increase, *p* = 0.028) and citrulline + arginine (235.7% increase, *p* = 6.3 × 10^−4^) enhanced total eNOS expression compared to controls. Neither citrulline (*p* = 0.22) nor arginine (*p* = 0.11) alone altered the expression of p-eNOS compared to controls, but when citrulline + arginine was added, p-eNOS (Ser1177) expression was significantly higher than controls (265.2% increase, *p* = 0.029). These data suggest citrulline + arginine stimulates the citrulline-NO cycle by activating eNOS via phosphorylation at Serine1177 to produce intracellular NO.

### 2.5. eNOS Inhibition Prevents Citrulline Plus Arginine-Induced Angiogenesis and Increases Permeability in HRMECs

To confirm the role of eNOS in citrulline + arginine-induced angiogenesis, HRMECs were treated with citrulline + arginine in the presence of the eNOS inhibitor, Nω-Nitro-L-arginine methyl ester hydrochloride (L-NAME). Citrulline + arginine induced cell proliferation (Figure 7A) compared to untreated controls (33.3% increase, *p* = 0.013), and this proliferation was not significantly affected by L-NAME (*p* = 0.17). There was a significant induction of cell migration (Figure 7B,C) in HRMECs treated with citrulline + arginine compared to untreated controls (39.6% increase, *p* = 5.4 × 10^−5^), and this induction of migration was inhibited in the presence of L-NAME compared to citrulline + arginine alone (17.7% decrease, *p* = 0.0080). Citrulline + arginine increased tube length (Figure 7D,E) compared to untreated controls (23.0% increase, *p* = 8.8 × 10^−5^), and L-NAME inhibited this increase in tube length compared to citrulline + arginine alone (11.9% decrease, *p* = 0.0092). These data show that eNOS activation is critical to the ability of citrulline + arginine to induce features of angiogenesis in retinal endothelial cells.

To determine the effect of eNOS in citrulline + arginine-induced permeability, HRMECs were treated with citrulline + arginine in the presence of L-NAME, and NO production (Figure 8A,B), FITC-dextran flow-through (Figure 8C), and TEER (Figure 8D) were measured. Citrulline + arginine enhanced NO production (123.5% increase, *p* = 0.0036) compared to untreated controls, and L-NAME inhibited this increase in NO production compared to citrulline + arginine alone (44.1% decrease, *p* = 0.0092). Citrulline + arginine increased FITC-dextran flow-through (106.2% increase, *p* = 3.6 × 10^−4^) compared to untreated controls, and L-NAME inhibited this increase in FITC-dextran flow-through compared to citrulline + arginine alone (57.2% decrease, *p* = 1.1 × 10^−4^). Citrulline + arginine caused decreased TEER readings (21.1% decrease, *p* = 3.9 × 10^−4^) compared to untreated controls, and L-NAME prevented these changes in TEER readings compared to citrulline + arginine alone (27.4% increase, *p* = 3.0 × 10^−4^). Together, these data indicate that eNOS is required for citrulline + arginine to induce NO production and permeability of retinal endothelial cell monolayers.

## 3. Discussion

In this study, we investigated how exogenous citrulline and arginine affect human retinal endothelial cells in culture. We found that citrulline + arginine can induce cell proliferation, cell migration, and tube formation in retinal endothelial cells and increase permeability of retinal cell monolayers. To investigate the mechanism by which citrulline + arginine promotes angiogenesis and increases permeability in HRMECs, we measured arginase activity, protein expression of both arginase isoforms, protein expression and phosphorylation of eNOS, and NO production. We found that citrulline + arginine increases eNOS expression and phosphorylation along with NO production, and that inhibiting eNOS blocks the angiogenic and permeability-increasing effects of citrulline + arginine. We found no effect of citrulline + arginine on arginase activity.

Our laboratory previously found elevated levels of citrulline and arginine in plasma of diabetic patients with DR compared to diabetic controls [5,6]. Arginine can be acted upon by arginase to produce urea or by NOS to produce NO and citrulline. Citrulline can act as an inhibitor of arginase [16], and exogenous citrulline has been shown to increase NO formation [16,17]. Altered arginase and NOS activity have both been linked to patients with DR [10,11,12,13], and in mouse models of DR, increased arginase activity was associated with decreased NO levels [11]. Therefore, we wished to determine whether citrulline + arginine promotes angiogenesis in HRMECs via arginase activation or through NOS activation and NO production.

Diabetes and high glucose models in vitro and in rodents have been associated with increased arginase activity and increased expression of both arginase isoforms, arginase-1 and arginase-2 [11,18,19,20]. Although both isoforms perform the same function in the urea cycle, the two isoforms differ in their cellular localization and contribution to retinal vascular pathology. Arginase-1 is found in the cytoplasm and has been shown in retinal ischemia/reperfusion (I/R) and oxygen-induced retinopathy (OIR) models in mice to protect against neurovascular degeneration and to limit vitreoretinal neovascularization [21,22]. Arginase-2 is located in the mitochondria and has been found to promote neurovascular degeneration in I/R and OIR mouse models [23,24] and to increase expression of inflammatory mediators and promote cell death in murine and in vitro diabetes models [18]. We have demonstrated here that citrulline and arginine, alone and in combination, did not significantly affect arginase activity in HRMECs. While previous studies demonstrated decreased arginase activity in cells treated with citrulline at concentrations of 300 µM to 2.5 mM [17,25], we found that adding citrulline at concentrations of 30 µM with 70 µM arginine to HRMECs decreased arginase-1 protein levels without affecting arginase-2 levels or overall arginase activity. The concentration of citrulline used in the present study may be too low to inhibit arginase activity but high enough to promote the citrulline-NO cycle in retinal endothelial cells in normal glucose conditions.

It was recently shown that arginase-1 knockout mice subjected to the OIR model displayed increased pathologic retinal neovascularization and increased retinal citrulline and arginine compared to wild-type OIR mice [22]. The same study showed that cell-specific deletion of endothelial arginase-1 caused reduced sprouting in a choroidal sprouting assay. This fits with our data showing that exogenous citrulline + arginine stimulates angiogenesis in retinal endothelial cells and decreases arginase-1 expression. It may be that the exogenous citrulline is primarily inhibiting arginase-1, which normally protects against pathologic neovascularization, and the arginine provides additional substrate for eNOS to produce NO, thus stimulating angiogenesis.

NO is produced when eNOS is activated by phosphorylation and converts arginine to citrulline [15]. Several activating phosphorylation sites are present on eNOS, including serine 1177, which is phosphorylated by VEGF and other factors [15]. Exogenous arginine has been shown to increase NOS activity [26], and citrulline has also been found to increase NO formation [16]. Citrulline + arginine has been shown to increase eNOS expression and phosphorylation and increase NO production in human umbilical venous endothelial cells (HUVEC) [25], consistent with our findings in the current study in retinal endothelial cells. We found that citrulline + arginine increases expression of total and serine 1177-phosphorylated eNOS and NO production in retinal endothelial cells in a manner similar to cells treated with VEGF. An increase in both eNOS and p-eNOS implies that the treatment of citrulline + arginine is causing increased transcription of eNOS, and this increased eNOS is being phosphorylated to create more active NO, and not just increased activation of eNOS. Future work is needed to further investigate this.

In vivo, overexpression of eNOS has been shown to stimulate retinal angiogenesis, and this effect is mitigated with the eNOS inhibitor, L-NAME [27]. Previous studies have also shown inhibition of VEGF-induced endothelial cell migration, proliferation, and sprouting using eNOS siRNA, and eNOS-deficient mice displayed decreased vascularization during development and decreased OIR induced neovascularization [28]. To determine if eNOS activation is required for citrulline + arginine to promote angiogenesis in retinal endothelial cells, we performed proliferation, migration, and tube formation assays, treating with citrulline + arginine in the presence and absence of the eNOS inhibitor L-NAME. We found that inhibiting eNOS blocked the ability of citrulline + arginine to induce cell migration and tube formation, providing further evidence that citrulline + arginine promotes angiogenesis through eNOS and NO production.

Increased eNOS has also been shown to be associated with retinal vascular permeability in STZ-induced diabetic rats [29], and this increased permeability was inhibited in OIR mice using the eNOS inhibitor, L-NMMA, to suppress NO formation [30]. Previous studies have investigated the importance of NO production in the ability of VEGF to increase vascular permeability [13,31]. Bovine retinal endothelial cells displayed decreased TEER readings and increased FITC-dextran flow through when exposed to VEGF [32], and such VEGF-induced permeability was inhibited by L-NAME in HUVEC in vitro [33] and guinea pigs in vivo [34]. To determine if eNOS-induced NO is required for citrulline + arginine to increase permeability in retinal endothelial cells, we measured permeability of retinal endothelial cell monolayers, treating with citrulline + arginine in the presence and absence of L-NAME. We found that inhibiting eNOS prevented citrulline + arginine-induced NO production and changes in the retinal endothelial cell monolayer, providing further evidence that citrulline + arginine increases endothelial permeability through eNOS and NO production.

While this study is limited by its use of in vitro experiments to investigate the effects of citrulline + arginine on retinal endothelial cells, we did use commercially available primary human retinal endothelial cells and three separate assays to study the effect of citrulline + arginine on angiogenesis, as well as two assays to evaluate endothelial cell monolayer permeability. We recognize that it is possible that different concentrations of citrulline and arginine may have yielded different experimental results, but we used the mean citrulline and arginine concentrations we previously found in plasma in an attempt to simulate biologically relevant concentrations.

In this study, we demonstrated that citrulline + arginine treatment induces an angiogenic response and activates eNOS to increase NO production in human retinal endothelial cells. Our in vitro findings suggest that citrulline + arginine may contribute to pathologic angiogenesis and increased retinal vascular permeability through eNOS upregulation. Future studies to determine if citrulline + arginine-induced angiogenesis and endothelial cell permeability are independent of the VEGF/VEGFR pathway will be important to determine if these metabolites may be targets for the prevention or treatment of retinal vascular diseases involving aberrant angiogenesis and vascular permeability.

## 4. Materials and Methods

### 4.1. Cell Culture

Primary human retinal microvascular endothelial cells (HRMEC)s were purchased from Cell Systems (ACBRI 181, Kirkland, WA, USA). HRMECs were cultured in EBM Endothelial Cell Growth Basal Medium, Phenol Red Free (Lonza, CC-3129, Basel, Switzerland) supplemented with 10% fetal bovine serum (FBS, Gibco, Carlsbad, CA, USA) and EGM endothelial cell growth medium SingleQuots (Lonza, CC-4133). Prior to experimental procedures for angiogenesis assays, HRMECs were serum-starved overnight in medium supplemented with 0.5% FBS. Experiments for angiogenesis were performed in medium with 0.5% FBS, and experiments for monolayer integrity were performed in medium with 10% FBS. The concentrations of citrulline (30 µM) and arginine (70 µM) used for experiments were determined by using the average plasma concentrations measured in patients with DR in our previous study [6]. In eNOS-blocking experiments, the eNOS inhibitor Nω-Nitro-L-arginine methyl ester hydrochloride (L-NAME) was used at 500 µM [27,35,36,37]. HRMECs were incubated at 37 °C, 5% CO_2_, 20.9% O_2_, and 95% relative humidity.

### 4.2. Proliferation Assay

HRMECs were seeded at a density of 1.5 × 10^4^ in triplicate in a 96-well plate. HRMECs were treated with 30 µM citrulline (Cit), 70 µM arginine (Arg), or Cit + Arg. Untreated cells were used as a negative control, and cells treated with 100 ng/mL vascular endothelial growth factor (VEGF) were used as a positive control. Cell proliferation was measured using the BrdU Proliferation Assay (Cell Signaling Technology #6813S, Danvers, MA, USA) following the manufacturer’s instructions. Briefly, HREMCs were treated for 5 h, and BrdU was added to the wells overnight. The BrdU ELISA was performed 24 h after the addition of citrulline + arginine. Absorbance readings at 450 nm from each well were normalized to the average absorbance value of the untreated control. Normalized absorbance values were compared to normalized absorbance of the control.

### 4.3. Scratch Wound Migration Assay

HRMECs were plated in 12-well plates and grown to confluency. A 200 µL pipette tip was used to create the scratch, an area devoid of cells in the middle of each well. HRMECs were treated with 30 µM Cit, 70 µM Arg, or Cit + Arg. Untreated cells were used as a negative control, and cells treated with 100 ng/mL VEGF were used as a positive control. Images of scratches were taken at 0 and 16 h after treatment. The areas of the scratches devoid of cells were measured using ImageJ (V1.53t, National Institute of Mental Health, Bethesda, MD, USA). The area of cell migration was calculated by subtracting the area of the scratch at 16 h from the area of the scratch at 0 h. All migration values were normalized to the migration value of untreated controls.

### 4.4. Tube Formation Assay

HRMECs were seeded onto 150 µL polymerized growth reduced Matrigel in 48-well plates at a density of 1 × 10^5^. HRMECs were treated with 30 µM Cit, 70 µM Arg, or Cit + Arg and added to Matrigel. Untreated cells served as the negative control, and cells treated with 100 ng/mL VEGF were the positive control. HRMECs were allowed to form tubes for 7 h, and images were taken from five standard fields of each well. Tube length was measured using ImageJ, and measurements for each sample were averaged.

### 4.5. Arginase Activity Assay

HRMECs were seeded onto 6-well plates and grown to confluency. Cells were treated with 30 µM Cit, 70 µM Arg, or Cit + Arg, and untreated cells served as the control. After 24 h, cells were washed with PBS and pelleted by centrifuging at 1000× *g* for 10 min. Supernatant was removed, and the cell pellet wasd resuspended and lysed in 10 mM Tris with 0.4% Tween20 supplemented with 1µM leupeptin and pepstatin A. The sample was centrifuged at 10,000× *g* for 10 min, and the resulting supernatant was used for the arginase activity assay (Sigma #MAK112, St. Louis, MO, USA) following the manufacturer’s instructions. Briefly, 20 µL of each sample and 20 µL water was added to a 96-well plate in duplicate, and one set of samples (sample) was incubated with fresh arginine buffer for 2 h at 37 °C. Substrate solution was added to all wells, and arginine buffer was added to the other set of samples (sample blank) for 1 h at room temperature. The plate was read at 430 nm, and arginase activity was calculated as described in the kit’s protocol with the equation below. Arginase activity was normalized to total protein concentration of each sample using a BCA assay (Pierce, ThermoFisher, Waltham, MA, USA).(1)$$Activity=\frac{{\left(A_{430}\right)}_{sample}-{\left(A_{430}\right)}_{blank}}{{\left(A_{430}\right)}_{standard}-{\left(A_{430}\right)}_{water}}\times \frac{1\;nM\times 50\times 10^{3}}{V\times T}$$

### 4.6. Detecting NO Levels in HRMECs Using DAF-FM DA

HRMECs were seeded onto chambered slides at a density of 2.5 × 10^4^. HRMECs were treated with 30 µM Cit, 70 µM Arg, or Cit + Arg. Untreated cells served as the negative control, and cells treated with 100 ng/mL VEGF were the positive control. After 24 h, cells were incubated with 10 µM DAF-FM DA for 30 min at 37 °C. Following incubation, cells were washed once with PBS, and fresh serum-free medium was added for 20 min at 37 °C. Cells were washed once more with PBS and mounted with Fluoromount-G (SouthernBiotech, Birmingham, AL, USA). Cells were imaged with a fluorescent microscope, and fluorescence intensity was measured using ImageJ.

### 4.7. FITC-Dextran Permeability Assay

HRMECs were plated at a density of 1.0 × 10^5^ on fibronectin coated Transwell inserts (0.4 µm) and grown for one week to form a monolayer. Cells were treated with 30 µM Cit, 70 µM Arg, or Cit + Arg for 24 h. FITC-dextran (25 ng/mL) was added in fresh medium to the inserts, and the medium in the wells was replaced. Two hours after adding FITC-Dextran, 50 µL from the well was diluted into 50 µL PBS in duplicate, and this flow-through was collected from the well and its fluorescence was measured at 490 nm and 520 nm and averaged for the samples.

### 4.8. Trans-Endothelial Electrical Resistance Assay

HRMECs were plated at a density of 1.0 × 10^5^ on fibronectin coated Transwell inserts (0.4 µm) and grown until a monolayer was achieved. Cells were treated with 30 µM Cit, 70 µM Arg, or Cit + Arg, and trans-endothelial electrical resistance (TEER) readings were collected at 0, 24, and 48 h after treatment from three standard areas of each sample using chopstick electrodes (EVOM2, World Precision Instruments, Sarasota, FL, USA) and averaged together. The average TEER reading of a blank insert was subtracted from each sample.

### 4.9. Western Blot

HRMECs were seeded onto 60 mm dishes and grown to 90% confluency. Cells were treated with 30 µM Cit, 70 µM Arg, or Cit + Arg for 2 h. Untreated cells served as a negative control. In angiogenesis and VEGF downstream pathway experiments, cells treated with 100 ng/mL VEGF were used as a positive control. After treatment, cells were washed once in cold PBS and collected in PBS. Cells were pelleted at 1000 rpm for 5 min and lysed in radioimmunoprecipitation assay (RIPA) buffer using sonication. Total protein concentration of samples was determined using a BCA Assay. Proteins were separated using 8% acrylamide gels and electrophoretically transferred to nitrocellulose membranes. Membranes were blocked with 1% BSA in ddH_2_O and were probed with antibodies against eNOS (Cell Signaling Technology #9572, Danvers, MA, USA), phosphorylated eNOS (Serine 1177) (Cell Signaling Technology #9570), arginase-1 (ProteinTech #16001-1-AP, Rosemont, IL, USA), or arginase-2 (Cell Signaling Technology #55003) at 1:1000 dilution in 5% BSA in TBS + 0.1% Tween20 (TBS-T) overnight at 4 °C. Membranes were washed 3 times with TBS-T, and secondary antibody goat anti-rabbit IgG (Millipore #12-348, Burlington, MA, USA) in 5% milk in TBS-T was added to the blots for 1 h at room temperature. After washing the membranes 3 times with TBS-T, the chemiluminescence substrate ECL (Pierce, ThermoFisher, Waltham, MA, USA) was added for development, and the membranes were imaged using ChemiDoc (Bio-Rad, Hercules, CA, USA). Band intensity was quantified using ImageJ. Beta-actin conjugated with HRP (1:10,000; Cell Signaling Technology #12262) was used as a loading control.

### 4.10. Statistical Analysis

Values from each assay were compared using a one-way ANOVA with Tukey’s Honest Significant Difference post hoc test. HRMECs treated with citrulline, arginine, citrulline + arginine, or VEGF were compared to untreated HRMECs controls. A *p*-value < 0.05 was considered significant for all assays.

## Figures and Tables

**Figure 1 ijms-26-02080-f001:**
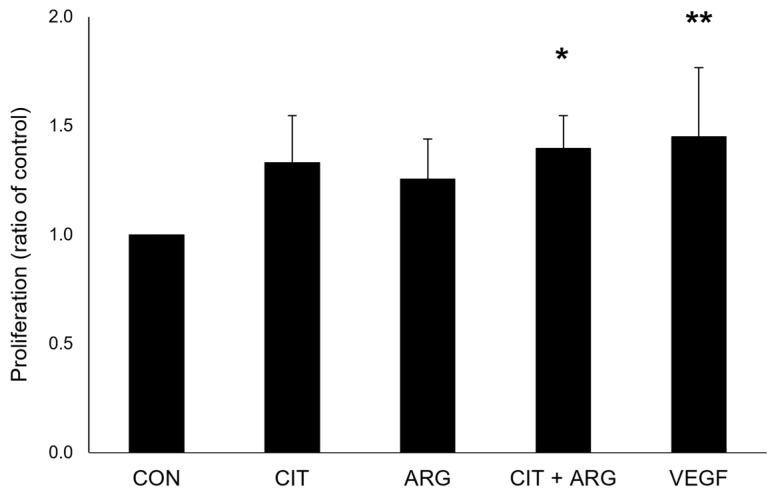
Citrulline + arginine induces cell proliferation in HRMECs. HRMECs were untreated (CON) or treated with 30 µM citrulline (CIT), 70 µM arginine (ARG), 30 µM citrulline + 70 µM arginine (CIT + ARG), or 100 ng/mL vascular endothelial growth factor (VEGF) for 24 h, and proliferation was measured using a BrdU proliferation assay. Absorbance values were normalized to average absorbance value of untreated controls. Normalized absorbance values were compared to normalized absorbance of the control. Data are presented as mean ± standard deviation (n = 6). * *p* < 0.05; ** *p* < 0.01 compared to control.

**Figure 2 ijms-26-02080-f002:**
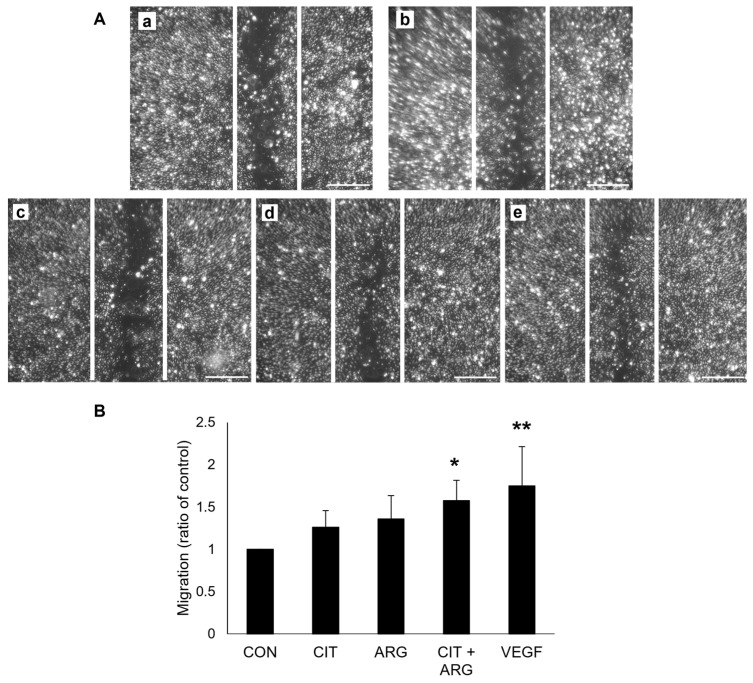
Citrulline + arginine induces cell migration in HRMECs. HRMECs were untreated (CON) or treated with 30 µM citrulline (CIT), 70 µM arginine (ARG), 30 µM citrulline + 70 µM arginine (CIT + ARG), or 100 ng/mL vascular endothelial growth factor (VEGF). Replicate scratches were applied to confluent cells, and HRMECs were treated for 16 h. (**A**) Representative scratch wound migration images of (**a**) untreated controls, (**b**) 100 ng/mL VEGF, (**c**) 30 µM citrulline, (**d**) 70 µM arginine, and (**e**) citrulline + arginine. White lines represent the initial scratch area. Scale bar, 500 µm. (**B**) Migration was measured as the difference in area of scratch at 16 h and area of scratch at 0 hr. All migration values were normalized to the migration value of untreated controls. Data are presented as mean ± standard deviation (n = 6). * *p* < 0.05; ** *p* < 0.01 compared to control.

**Figure 3 ijms-26-02080-f003:**
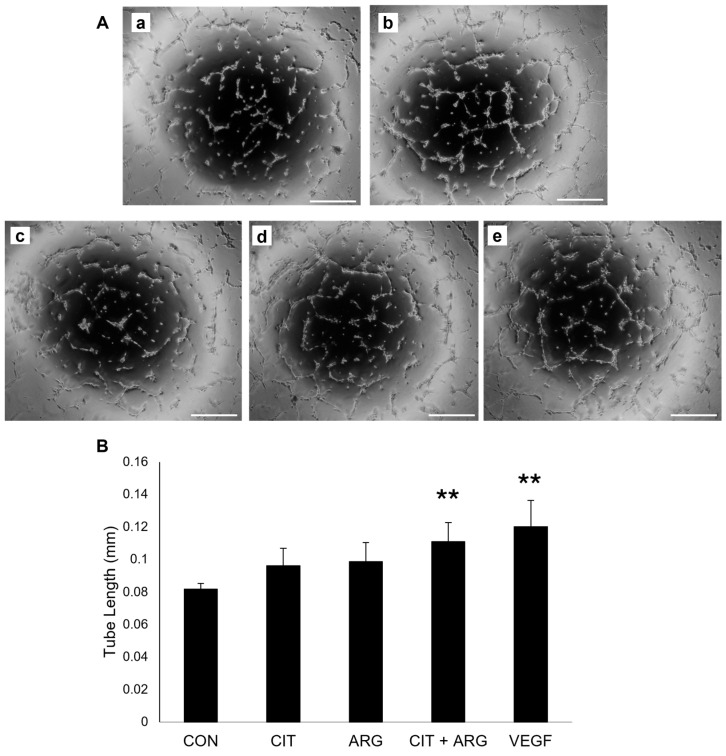
Citrulline + arginine induces tube formation in HRMECs. HRMECs were untreated (CON) or treated with 30 µM citrulline (CIT), 70 µM arginine (ARG), 30 µM citrulline + 70 µM arginine (CIT + ARG), or 100 ng/mL vascular endothelial growth factor (VEGF) and seeded onto growth reduced Matrigel for 6 h. (**A**) Representative tube formation images of (**a**) untreated controls, (**b**) 100 ng/mL VEGF, (**c**) 30 µM citrulline, (**d**) 70 µM arginine, and (**e**) citrulline + arginine. Scale bar, 500 µm. (**B**) Tube length of each sample was measured from five standard fields of each sample and averaged. Data are presented as mean ± standard deviation (n = 6). ** *p* < 0.01 compared to control.

**Figure 4 ijms-26-02080-f004:**
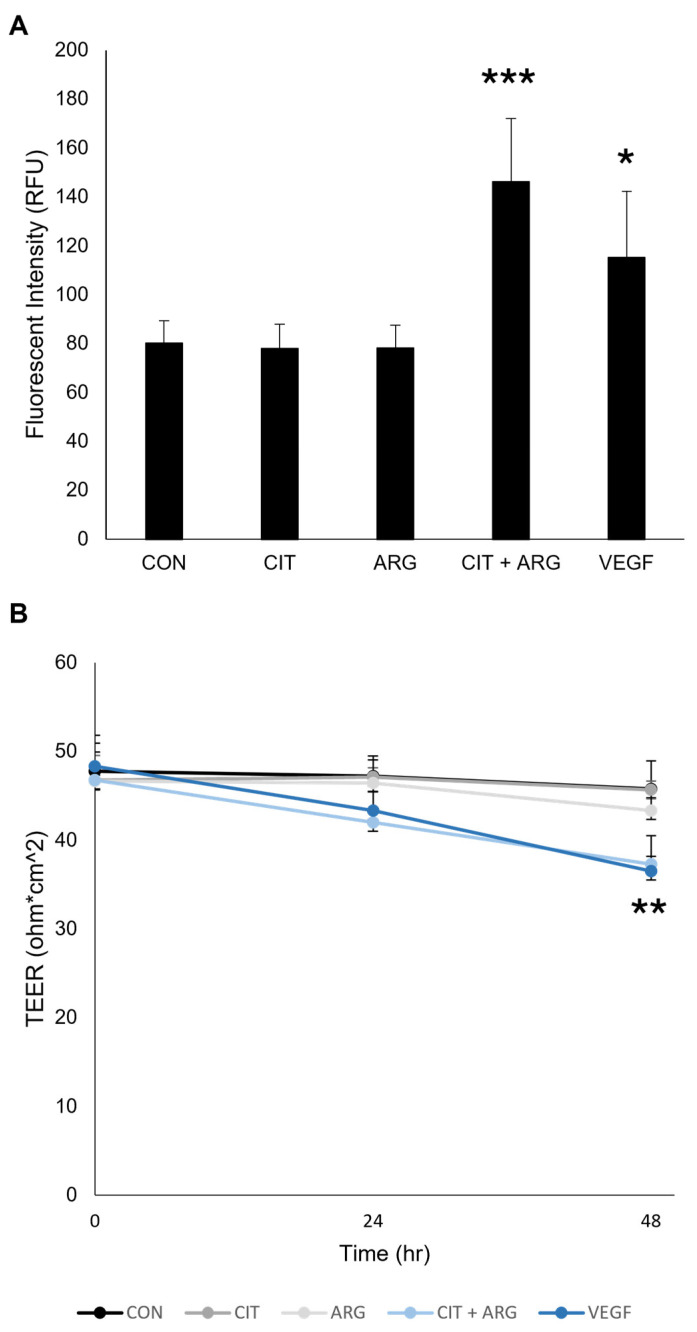
Citrulline + arginine increases permeability of HRMEC monolayers. HRMECs were grown on Transwell inserts for a week to form a monolayer and untreated (CON) or treated with 30 µM citrulline (CIT), 70 µM arginine (ARG), 30 µM citrulline + 70 µM arginine (CIT + ARG), or 100 ng/mL vascular endothelial growth factor (VEGF). (**A**) After 24 h, FITC-Dextran was added to the insert, the flow-through was collected from the well and measured (n = 6). (**B**) HRMEC TEER was measured at 0, 24, and 48 h after treatment (n = 3). Data are presented as mean ± standard deviation. * *p* < 0.05, ** *p* < 0.01, *** *p* < 0.001 compared to untreated controls.

**Figure 5 ijms-26-02080-f005:**
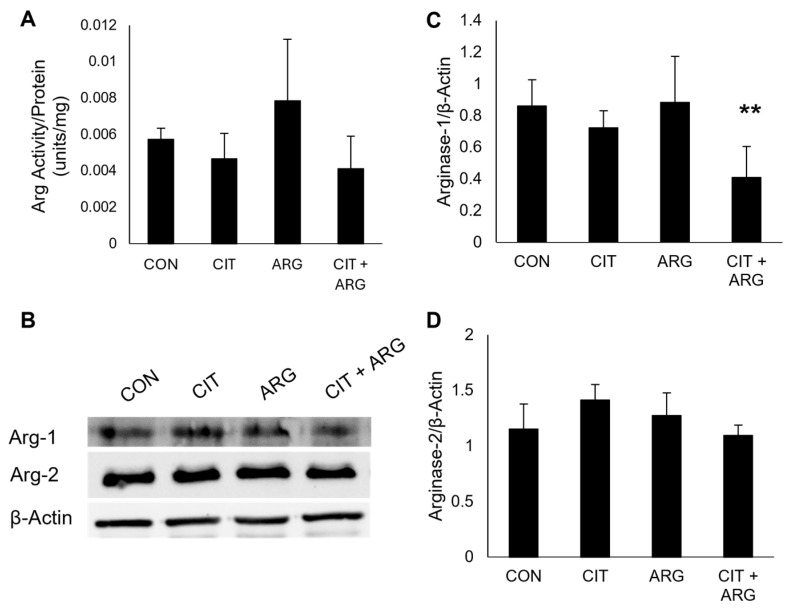
Citrulline + arginine does not affect arginase activity or arginase-2 expression but leads to lower arginase-1 expression in HRMECs. HRMECs were untreated (CON) or treated with 30 µM citrulline (CIT), 70 µM arginine (ARG), or 30 µM citrulline + 70 µM arginine (CIT + ARG). (**A**) HRMECs were treated for 24 h, and cell lysates were collected for a colorimetric arginase activity kit to convert byproducts of arginase to urea. Protein concentration of lysates was determined with BCA assay, and arginase activity values were normalized to total protein of sample. Data are presented as mean ± standard deviation (n = 6). (**B**–**D**) HRMECs were treated for 2 h, and cell lysates were collected for Western blot. (**B**) Representative Western blots of arginase-1 and arginase-2 protein expression. Quantification of (**C**) arginase-1 and (**D**) arginase-2. Data are presented as mean ± standard deviation (n = 6). ** *p* < 0.01 compared to control.

**Figure 6 ijms-26-02080-f006:**
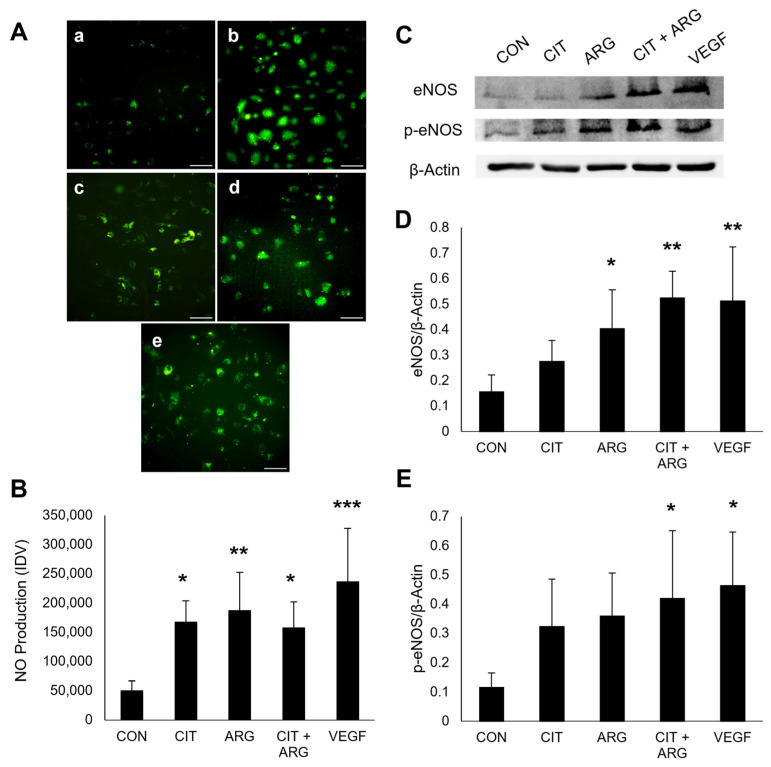
Citrulline + arginine induces NO production and increases eNOS expression and phosphorylation in HRMECs. HRMECs were untreated (CON) or treated with 30 µM citrulline (CIT), 70 µM arginine (ARG), 30 µM citrulline + 70 µM arginine (CIT + ARG), or 100 ng/mL vascular endothelial growth factor (VEGF). (**A**) Representative images of NO production in (**a**) untreated controls, (**b**) 100 ng/mL VEGF, (**c**) 30 µM citrulline, (**d**) 70 µM arginine, and (**e**) 30 µM citrulline + 70 µM arginine. Scale bar, 100 µm. (**B**) Quantification of NO production using ImageJ (V1.53t). Data are presented as mean ± standard deviation (n = 6). * *p* < 0.05; ** *p* < 0.01; *** *p* < 0.001 compared to control. (**C**) Representative blots of eNOS and p-eNOS. Quantification of (**D**) eNOS and (**E**) p-eNOS. Data are presented as mean ± standard deviation (n = 6). * *p* < 0.05 and ** *p* < 0.01 compared to control.

**Figure 7 ijms-26-02080-f007:**
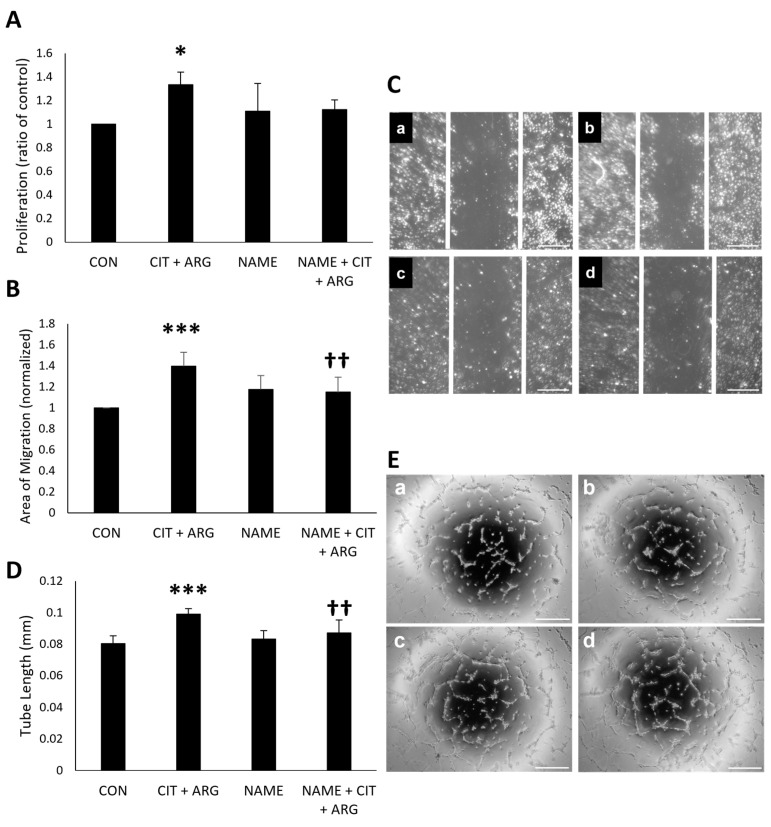
Citrulline + arginine-induced cell migration and tube formation are inhibited in the presence of L-NAME. HRMECs were untreated (CON) or treated with 30 µM citrulline + 70 µM arginine (CIT + ARG), 500 µM L-NAME (NAME), or 500 µM L-NAME with 30 µM citrulline + 70 µM arginine (NAME + CIT + ARG). (**A**) Proliferation was measured after 24 h, and absorbance values were normalized to average absorbance value of untreated controls. Normalized absorbance values were compared to normalized absorbance of the control. Data are presented as mean ± standard deviation (n = 6). (**B**) Migration data were collected after 16 h, and all migration values were normalized to the migration value of untreated controls. Data are presented as mean ± standard deviation (n = 6). (**C**) Representative images of migration in (**a**) untreated controls, (**b**) 30 µM citrulline + 70 µM arginine, (**c**) 500 µM L-NAME, and (**d**) 500 µM L-NAME + 30 µM citrulline + 70 µM arginine. White lines represent the initial scratch area. Scale bar, 500 µm. (**D**) Tube length was measured after 6 h, and data are presented as average tube length ± standard deviation (n = 6). (**E**) Representative images of tube formation in (**a**) untreated controls, (**b**) citrulline + arginine, (**c**) L-NAME, and (**d**) L-NAME + citrulline + arginine. Scale bar, 500 µm. For all graphs, * *p* < 0.05 and *** *p* < 0.001 compared to untreated control. †† *p* < 0.01 compared to cells treated with citrulline + arginine.

**Figure 8 ijms-26-02080-f008:**
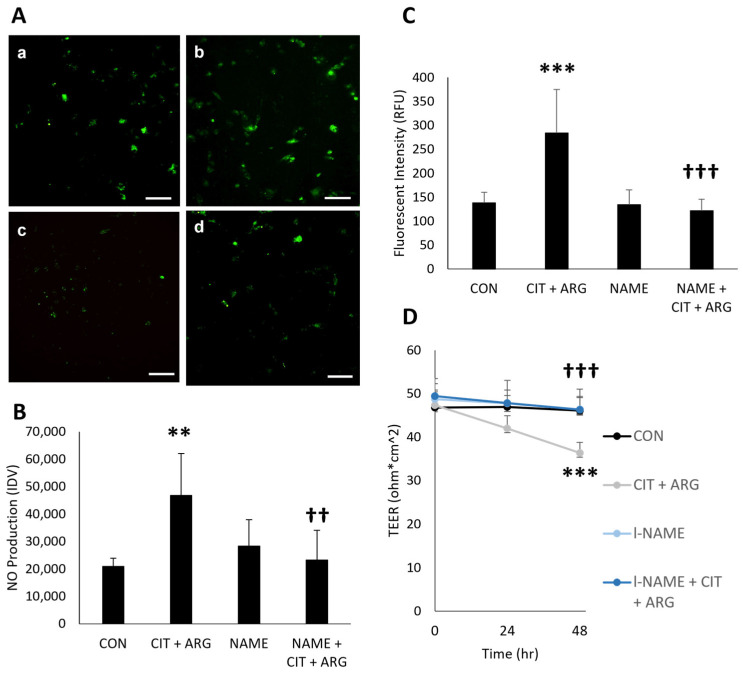
Citrulline + arginine-induced increase in HRMEC monolayer permeability is inhibited in the presence of L-NAME. HRMECs were untreated (CON) or treated with 30 µM citrulline + 70 µM arginine (CIT + ARG), 500 µM L-NAME (NAME), or 500 µM L-NAME with 30 µM citrulline + 70 µM arginine (NAME + CIT + ARG). (**A**) Representative images of NO production in (**a**) untreated controls, (**b**) 30 µM citrulline + 70 µM arginine, (**c**) 500 µM L-NAME and (**d**) 500 µM L-NAME + 30 µM citrulline + 70 µM arginine. Scale bar, 100 µm. (**B**) Quantification of NO production using ImageJ. Data are presented as mean ± standard deviation (n = 6). (**C**,**D**) HRMECs were grown on Transwell inserts for a week to form a monolayer and treated with 30 µM citrulline + 70 µM arginine (CIT + ARG), 500 µM L-NAME (NAME), or 500 µM L-NAME with 30 µM citrulline + 70 µM arginine (NAME + CIT + ARG). (**C**) After 24 h, FITC-Dextran was added to the insert, the flow-through was collected from the well and measured (n = 6). (**D**) HRMEC TEER was measured at 0, 24, and 48 h after treatment. Data are presented as mean ± standard deviation (n = 6). For all graphs, ** *p* < 0.01 *** *p* < 0.001 compared to untreated controls. †† *p* < 0.01 and ††† *p* < 0.001 compared to cells treated with citrulline + arginine.

## Data Availability

The original contributions presented in this study are included in the article. Further inquiries can be directed to the corresponding author.

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
