# Peer review of "Citrulline Plus Arginine Induces an Angiogenic Response and Increases Permeability in Retinal Endothelial Cells via Nitric Oxide Production"

_ijms, 2025, doi:10.3390/ijms26052080_

Round 1

Reviewer 1 Report

Comments and Suggestions for Authors

This is an excellent research paper on citrulline and arginine effects, inducing angiogenesis. This is an important issue considering diabetic retinopathy. The paper can be accepted for publication, after minor revision.

1. One of the main things is the Introduction section. Half of the short Introduction is the summary of the results, which is not appropriate. The authors should delete the results, only mention the aims of the research, and expand the Introduction with "real" Introduction, more background information on the topic.

2. The graphs are not uniformly presented, there is no y bar shown in most figures, while in some it is present.

3. Abbreviations in the figure legends should be completed, for example "VEGF (VEGF)" does not make sense, the whole name should be indicated in the legends. Also, in almost all figure legends, concetrnation are indicated for arg and cit, but not arg+cit together. please complete this information in the legends. The abbreviation "CON" should also be clearly indicated in the legends (control (CON)).

4. Figure 6, the western blots are very smeared, how did the authors use these smeared images for quantification?

5. Figure 5, the western blot images for Arg-1 are very blurry, could you replace it with better images?

Author Response

This is an excellent research paper on citrulline and arginine effects, inducing angiogenesis. This is an important issue considering diabetic retinopathy. The paper can be accepted for publication, after minor revision. 

Comment 1: One of the main things is the Introduction section. Half of the short Introduction is the summary of the results, which is not appropriate. The authors should delete the results, only mention the aims of the research, and expand the Introduction with "real" Introduction, more background information on the topic.  

Response 1: Thank you for the suggestion. We have revised the Introduction section to exclude the summary of the results and only include the aims of the research. We have also expanded on the Introduction section to include more background information on diabetic retinopathy and the significance of current study in the manuscript at Lines 30-68. 

Comment 2: The graphs are not uniformly presented, there is no y bar shown in most figures, while in some it is present. 

Response 2: Thank you for the suggestion. We have added the y axis to all bar graph figures. 

Comment 3: Abbreviations in the figure legends should be completed, for example "VEGF (VEGF)" does not make sense, the whole name should be indicated in the legends. Also, in almost all figure legends, concentration are indicated for arg and cit, but not arg+cit together. please complete this information in the legends. The abbreviation "CON" should also be clearly indicated in the legends (control (CON)). 

Response 3: Thank you for the suggestions. We have revised all figure captions to include the full abbreviations for VEGF and CON, as well as added the concentrations for ARG + CIT together and NAME + ARG + CIT together.  

Comment 4: Figure 6, the western blots are very smeared, how did the authors use these smeared images for quantification? 

Response 4: The western blots used in Figure 6 were only 1 of 6 replicates done. The images were quantified using the ImageJ software. While these images are not the clearest, we chose these western blot images specifically because they are the most representative of the quantification of the data. 

Comment 5: Figure 5, the western blot images for Arg-1 are very blurry, could you replace it with better images? 

Response 5: The reviewer suggests replacing the Arg-1 western blot images with ones that are less blurry. We understand their concern for the clarity of images; however, we chose these western blot images specifically because they are the most representative of the quantification of the data. We had an n = 6 so there were bound to be some outliers, as can be seen with the error bars, so we didn’t want to confuse the readers with a clearer western blot image that is not as representative of the entire data set.  

Reviewer 2 Report

Comments and Suggestions for Authors

The authors studied how exogenous citrulline and arginine affect human retinal endothelial cells in vitro culture. They observed that citrulline combined with arginine can promote retinal endothelial cell proliferation, migration, tube formation and permeability. They confirmed the combination increases eNOS and NO productions which cause angiogenesis. The in vitro experiments are nicely done. It would be more interesting if the authors declare by which pathways the exogenous citrulline and arginine regulate eNOS/NO productions in retinal endothelial cells.

Major concerns: Most of the studies are descriptive. some functional assays or in vivo models would enhance the quality of the manuscript. E.g. whether citrulline combined with arginine affect leukocyte transmigration crossing the pretreated retinal endothelial cells in vitro assay, or leukocyte trafficking in vivo system. Since the endothelial permeability is also affected by the treatment.

In the discussion, the authors point out the possible involvement of VEGF/VEGFRs pathway, while there is no direct evidence showing how the citrulline and arginine link to eNOS/NO regulation through VEGFRs except the similar induction of eNOS/NO. As a precursor to NO, it is not new that arginine alone promotes cell migration, tube formation, and endothelial dysfunction.  Since citrulline can convert to arginine, what it meaningful to combine these two?

Minor concerns:

1.    Fig.3a and 7E: tube formation showed no significantly difference among treated groups.

2.     Y axis is missing in most of the figures.

3.    Fig.6C: It is problematic that eNOS total and phosphorylation showed the same pattern, is the p-eNOS a real matter for function? If so, how it functions, which need to be explained in the text.

Author Response

The authors studied how exogenous citrulline and arginine affect human retinal endothelial cells in vitro culture. They observed that citrulline combined with arginine can promote retinal endothelial cell proliferation, migration, tube formation and permeability. They confirmed the combination increases eNOS and NO productions which cause angiogenesis. The in vitro experiments are nicely done. It would be more interesting if the authors declare by which pathways the exogenous citrulline and arginine regulate eNOS/NO productions in retinal endothelial cells. 

Major concerns:  

Comment 1: Most of the studies are descriptive. some functional assays or in vivo models would enhance the quality of the manuscript. E.g. whether citrulline combined with arginine affect leukocyte transmigration crossing the pretreated retinal endothelial cells in vitro assay, or leukocyte trafficking in vivo system. Since the endothelial permeability is also affected by the treatment. 

Response 1: The reviewer suggests confirming these results in an animal model or including some functional assays. We agree that these types of experiments would be great “next steps” in this line of research. However, we believe that we have conveyed a considerable amount of new information to the reader with the current studies. We do not have the funds or the resources to be able to perform additional experiments at this time. We address this limitation in the manuscript at Lines 373-377. 

Comment 2: In the discussion, the authors point out the possible involvement of VEGF/VEGFRs pathway, while there is no direct evidence showing how the citrulline and arginine link to eNOS/NO regulation through VEGFRs except the similar induction of eNOS/NO. As a precursor to NO, it is not new that arginine alone promotes cell migration, tube formation, and endothelial dysfunction.  Since citrulline can convert to arginine, what it meaningful to combine these two? 

Response 2: We did mention VEGF and the VEGF/VEGFR pathway. Of course, we used VEGF as a positive control for many of the experiments in the study since we know of its ability to cause endothelial cell angiogenesis and permeability. We also believe it will be very important for future studies to determine whether citrulline and arginine can promote NO production independent of VEGF/VEGF. We did not study that in this manuscript and draw no conclusions about it. If it turns out the citrulline and arginine can promote angiogenesis and permeability completely independent of the VEGF pathway, it may be that additional treatments targeting excess citrulline and arginine (or more specifically eNOS/NO) could serve as adjunct therapy with current anti-VEGF therapy. We did not discuss this in detail because it was not studied in this work, but we have clarified our referral to VEGF in the last paragraph of the Discussion at Lines 384-388.  

We studied the combination of citrulline and arginine because we found both molecules elevated in patients with diabetic retinopathy, and because previous studies have actually shown a wide range of effects of arginine on endothelial cells and in regard to diabetes and diabetic retinopathy. Studies typically evaluate the arginase pathway OR the nitric oxide pathway, but not both. We did not feel that one or the other pathway should be examined in isolation 

Minor concerns: 

Comment 1: Fig.3a and 7E: tube formation showed no significantly difference among treated groups. 

Response 1: The reviewer is concerned with Figures 3a and 7E showing no significant differences among the treated groups. Note that for our quantification, we are measuring tube length, not tube volume or quantity, so we understand that it might be hard to distinguish this solely based on the picture. That is why we have provided the quantification of the tube lengths using the ImageJ software.  

Comment 2: Y axis is missing in most of the figures.  

Response 2: Thank you for the suggestion. We have added the y axis to all bar graph figures. 

Comment 3: Fig.6C: It is problematic that eNOS total and phosphorylation showed the same pattern, is the p-eNOS a real matter for function? If so, how it functions, which need to be explained in the text.  

Response 3: The reviewer is wondering if it makes sense that total eNOS and phosphorylated-eNOS show the same (increasing) pattern. An increase in both eNOS and p-eNOS implies that the addition of exogenous citrulline + arginine is causing increased transcription of eNOS and that eNOS is being phosphorylated at a healthy (or increased) rate at the same time to create more NO. Future work is needed to further investigate this. We have addressed this in the Discussion section of the manuscript at Lines 344-347. 

Reviewer 3 Report

Comments and Suggestions for Authors The manuscript (citrulline plus arginine induces an angiogenic response and increases permeability in retinal endothelial cells via nitric oxide production) is interesting. However, several concerns should be resolved.

Each compound's dose dependency has not been clearly set. How 30 uM or 70 uM has been decided should be experimentally discussed. 

Basal medium's composition and information could be presented.

Migration condition should exclude the proliferation aspect. Control's condition should be clearly discussed.

Figure 6's western blot quality is poor. It should be replaced without bubble or crack of the bands.

It seems each image has no scale bars. It can be added.

Using one cell line is limited to make a clear conclusion. It needs several cell lines to generalize the conclusion.

For the method section, each quantification's description is not convincing. How the data is made should be clearly presented for each experiment. 

Although data description says n = 6, control has no standard deviation. It might need rationale how the data are organized. 

Author Response

The manuscript (citrulline plus arginine induces an angiogenic response and increases permeability in retinal endothelial cells via nitric oxide production) is interesting. However, several concerns should be resolved. 

Comment 1: Each compound's dose dependency has not been clearly set. How 30 uM or 70 uM has been decided should be experimentally discussed. 

Response 1: The reviewer questions how we determined the concentrations to use for our experiments and suggests performing dose-response experiments. Our lab previously measured concentrations of citrulline and arginine in plasma of patients with diabetic retinopathy compared to diabetic controls, and we found an increase of citrulline and arginine in patients with diabetic retinopathy. We used the mean concentration of citrulline and arginine from the cohort of patients with diabetic retinopathy from that study to perform the in vitro studies reported here. We explained this in the manuscript at Lines 377-380 and also at Lines 398-400. 

Comment 2: Basal medium's composition and information could be presented. 

Response 2: We have included the complete name of the basal medium we used, as well as the catalog number of the product on the Lonza website. We explain this in the manuscript at Lines 392-393. 

Comment 3: Migration condition should exclude the proliferation aspect. Control's condition should be clearly discussed.  

Response 3: We understand the reviewer’s question, in that if there is significant cell proliferation caused by a treatment, this could plausibly affect number of cells available to migrate. However, for this assay, we used a standard, well-cited protocol for the scratch-wound migration assay which measures migration at 16 hours, whereas proliferation in the BrdU assay is assayed at 24 hours. Given the time it takes for proliferation to occur, it is unlikely that new cells could for and have time to move into the scratch wound. 

Comment 4: Figure 6's western blot quality is poor. It should be replaced without bubble or crack of the bands.  

Response 4: The reviewer suggests replacing Figure 6’s western blot images with ones that are of better quality. We understand their concern for the clarity of images; however, we chose these western blot images specifically because they are the most representative of the quantification of the data. We had an n = 6 so there was bound to be some outliers, as can be seen with the error bars, so we didn’t want to confuse the readers with a clearer western blot image that is not as representative of the entire data set. 

Comment 5: It seems each image has no scale bars. It can be added. 

Response 5: Thank you for the suggestion. We have added scale bars to all necessary images. The description of the scale bars can be found in the captions of each image.  

Comment 6: Using one cell line is limited to make a clear conclusion. It needs several cell lines to generalize the conclusion.  

Response 6: The reviewer suggests including several cell lines to generalize our conclusions. While we agree that this would be valuable, we do not currently have the funds or the resources to be able to perform these experiments. We address this limitation in the manuscript at Lines 373-377. 

Comment 7: For the method section, each quantification's description is not convincing. How the data is made should be clearly presented for each experiment.  

Response 7: Thank you for the suggestions. We have edited the Materials and Methods section to provide more description for how each assay was quantified, including adding the equation that was used to calculate arginase activity.  

Comment 8: Although data description says n = 6, control has no standard deviation. It might need rationale how the data are organized.  

Response 8: The reviewer suggests including a rationale for the lack of standard deviation in the control data in Figures 1, 2B, 7A, and 7B. We want to point out that in these figures, the data were normalized to the control value. That is why the control data set does not have standard deviation/error bars. This is mentioned in the captions of Figure 1 (Lines 92-93) and Figure 2B (Lines 110-111); for Figure 7A and 7B, this rationale was added to the figure caption (Lines 247-252). 

Round 2

Reviewer 2 Report

Comments and Suggestions for Authors

This reviewer has no further comment. But the y axis in some figures is still beyond the standards. Some formal writing needed: e.g. "Citrulline + arginine", the "+" is not a word.

Author Response

Comment: This reviewer has no further comment. But the y axis in some figures is still beyond the standards. Some formal writing needed: e.g. "Citrulline + arginine", the "+" is not a word.

Response: We have added the y axis to all bar graph figures, and we agree with the reviewer that this helps present our data more clearly.  

We chose to write “citrulline + arginine” to show that this treatment group refers to a specific combination of the two metabolites. We didn’t want to confuse the readers when we listed our treatment groups since our cells were being treated with citrulline and arginine individually as well. After careful consideration, we chose to write “+” rather than “plus” for clarity to convey that the treatment is acting as a single component.  

Reviewer 3 Report

Comments and Suggestions for Authors

Basically, the raised concerns and comments are partly addressed. As a main issue, in vitro experiments' outcomes are easily changed by various conditions as far as I know and the authors may know either. Therefore, several cell types or lines with various doses of drugs and time of incubations are highly recommended to make a solid conclusion of the topic. This will make the current manuscript have a clear impact. However, it seems that it is difficult to cover this based on the authors' talk/answer. Hope it will be done near the future. 

Author Response

Comment: Basically, the raised concerns and comments are partly addressed. As a main issue, in vitro experiments' outcomes are easily changed by various conditions as far as I know and the authors may know either. Therefore, several cell types or lines with various doses of drugs and time of incubations are highly recommended to make a solid conclusion of the topic. This will make the current manuscript have a clear impact. However, it seems that it is difficult to cover this based on the authors' talk/answer. Hope it will be done near the future. 

Response: We understand and agree with the reviewers that confirming these results in several cell types or lines would be great “next steps” in this line of research. However, as mentioned in the first round of comments, we do not have the funds or the resources to be able to perform additional experiments at this time. Still, we believe that we have conveyed a considerable amount of new information to the reader with the current studies. We hope to inspire new research with this publication that will take these next steps to build on the work we have done in the current study.